

# Massless Rarita-Schwinger equations:
# Half and three halves spin solution

**Mauricio Valenzuela⋆ and Jorge Zanelli**

Centro de Estudios Científicos (CECs), Arturo Prat 514, Valdivia, Chile
Facultad de Ingeniería, Arquitectura y Diseño, Universidad San Sebastián, Valdivia, Chile

⋆ mauricio.valenzuela@uss.cl

## Abstract

Counting the degrees of freedom of the massless Rarita-Schwinger theory is revisited using Behrends-Fronsdal projectors. The identification of the gauge invariant part of the vector-spinor is thus straightforward, consisting of spins $\frac{1}{2}$ and $\frac{3}{2}$. The validity of this statement is supported by the explicit solution found in the standard gamma-traceless gauge. Since the obtained systems are deterministic –free of arbitrary functions of time– we argue that the often-invoked residual gauge symmetry lacks fundamental grounding and should not be used to enforce new external constraints. The result is verified by the *total Hamiltonian* dynamics. We conclude that eliminating the spin-$\frac{1}{2}$ mode *via* the *extended Hamiltonian* dynamics would be acceptable if the Dirac conjecture was assumed; however, this framework does not accurately describe the original Lagrangian system.

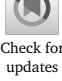

# 1 Introduction

In supergravity, the kinetic term for the gravitino field is the so-called Rarita-Schwinger (**RS**) Lagrangian (see, e. g., [1–3]),

$$\mathcal{L}[\psi] := -\frac{i}{2}\bar{\psi}_\mu \gamma^{\mu\nu\lambda}\partial_\nu\psi_\lambda \,. \tag{1}$$

This is not, however, the Lagrangian originally proposed by Rarita and Schwinger [4], as pointed out *e.g.*, in Weinberg's text [5].

Indeed, the study of relativistic quantum fields of arbitrary spin $s > 1$ was pioneered by Dirac [6], and followed by Fierz and Pauli [7]. Subsequently, Rarita and Schwinger greatly simplified the approach by describing a spin $l + 1/2$ field employing a wave function that was a spinor and a tensor of rank $l$, $\psi^\alpha_{\mu_1\cdots\mu_l}$, symmetric in its spacetime indices $(\mu_1\cdots\mu_l)$ satisfying a massive Dirac equation,

$$(\partial\!\!\!/ + M)\psi_{\mu_1\mu_2\cdots\mu_l} = 0 \,, \tag{2}$$

with the additional Lorentz-group irreducibility condition

$$\gamma^\mu \psi_{\mu\mu_2\cdots\mu_k} = 0 \,. \tag{3}$$

If $\psi_{\mu_1\mu_2\cdots\mu_l}$ is a solution of (2) and (3) with $M \neq 0$, then it also necessarily satisfies

$$\psi^\mu{}_{\mu\mu_2\cdots\mu_l} = 0 \,, \tag{4}$$
$$\partial^\mu\psi_{\mu\mu_2\cdots\mu_l} = 0 \,. \tag{5}$$

The simplest nontrivial RS system is the spin–$\frac{3}{2}$ field ($l = 1$) and the field $\psi^\alpha_\mu$ is a vector-spinor, for which the condition (4) is removed. Rarita and Schwinger [4] mention the existence of a family of Lagrangians parametrized by the mass ($M$) and a dimensionless coefficient ($A$) that gives rise to the equations (2) and (3). In [8–10] (for recent uses see [11]) such a family is proposed for $D = 4$ in the form

$$
\mathcal{L}_{(M,A)} := \frac{i}{2}\Big(\bar{\psi}^\mu(\partial\!\!\!/ + M)\psi_\mu + A\bar{\psi}^\mu(\gamma_\mu\partial_\nu + \gamma_\nu\partial_\mu)\psi^\nu + \frac{1}{2}(3A^2 + 2A + 1)\psi^\mu\gamma_\mu\partial\!\!\!/\gamma_\nu\psi^\nu
$$
$$
- M(3A^2 + 3A + 1)\psi^\mu\gamma_\mu\gamma_\nu\psi^\nu\Big). \tag{6}
$$

The dimensionless constant $A$ can take arbitrary values, but for $A = -1/2$ condition (3) with $l = 1$ does not follow from (2). The RS Lagrangian [4] corresponds to $A = -1/3$, "which permits a relatively simple expression of the equations of motion in the presence of electromagnetic fields."

With the advent of supergravity in the 1970s [12, 13], the community regularly referred to the massless limit of (6) with $A = -1$, which coincides with (1), as "the Rarita-Schwinger Lagrangian." The relativistic equation obtained from (1), often referred to as the RS equation (see *e.g.* [2, 3, 5]), is[1]

$$\gamma^{\mu\nu\lambda}\partial_\nu\psi_\lambda = 0 \,. \tag{7}$$

Note, however, that the massless limit of (2)-(3) is not equivalent to (7), and the variation of the massless Lagrangian (1) does not imply the irreducibility condition $\gamma^\mu\psi_\mu = 0$. This means that the dynamical contents of (2)-(3), for $l = 1$, in the limit $M \to 0$ are not guaranteed to be exactly the same as those of Lagrangian (1).

---

[1]Here $\gamma_\mu$, $\{\gamma_\mu,\gamma_\nu\} = 2\eta_{\mu\nu}$, are Dirac matrices, $\eta_{\mu\nu} = \mathtt{diag}(-1, 1, \dots)$ and $\gamma_{\mu\cdots\nu} = \gamma_{[\mu}\cdots\gamma_{\nu]}$ are completely antisymmetric products. We assume the Majorana reality condition $\psi^\dagger = \psi$, $\bar{\psi} = \psi^t C$, were $C^t = -C$.

## Gauge symmetry

The Lagrangian (1) changes by a total derivative under the gauge transformation,

$$\delta\psi_\mu = \partial_\mu\epsilon\,, \tag{8}$$

consequently, the field equations (7) are invariant under (8). Equation (2), on the other hand, does not exhibit a similar local invariance for an arbitrary local spinor parameter. In fact, Rarita and Schwinger observed that "in the exceptional case of zero rest mass, the wave function admits a gauge transformation given by (8)" with the additional requirement $\slashed{\partial}\epsilon = 0$. They saw this as a curious symmetry in an exceptional case but did not emphasize the gauge invariance of the action. In the case $A = -1$, the condition $\slashed{\partial}\epsilon = 0$ is unnecessary, and the gauge invariance under (8) is a true (off-shell) symmetry of the action.

Using properties of the Dirac matrices, equation (7) can also be written as

$$\slashed{\partial}\psi_\mu - \partial_\mu(\gamma\cdot\psi) = 0\,, \tag{9}$$

while

$$\slashed{\partial}(\gamma\cdot\psi) - \partial\cdot\psi = 0\,, \tag{10}$$

appears as a consistency condition.

It is a textbook argument that the gauge symmetry (8) can be used to choose the ($\gamma$-traceless) gauge $\gamma^\mu\psi_\mu = 0$, which together with (9) become equivalent to,

$$\slashed{\partial}\psi_\mu = 0\,, \qquad \gamma\cdot\psi = 0\,, \tag{11}$$

and we obtain that

$$\partial\cdot\psi = 0\,, \tag{12}$$

as a consistency condition. Gauge conditions are necessary to eliminate the ambiguity in time evolution, characterized by redundancies in solution space known as gauge orbits. As we shall see, the set of equations (11)-(12) is deterministic: once the initial conditions are specified, the field equations uniquely determine its evolution. An the explicit solution is given in (46) with the vector-spinor containing both spins, half and three halves. Consequently, introducing additional external conditions is unnecessary.

Based on the similarity between (12) and the Lorenz gauge in electromagnetism, one might be led to think that the divergence-free and the gamma-trace constraints are in the same—gauge-fixing type—footing and that they remove $2k$ degrees of freedom,[2] the equivalent of two spinor fields. In this logic, the system contains no spin–$\frac{1}{2}$ sector. This statement, however, is incorrect. The gauge freedom (8) is insufficient to set $\psi_\mu$ both gamma-traceless and divergence-free simultaneously starting from a generic off-shell configuration since there is only one spinor gauge-parameter to be used.

The consistency condition (12) is guaranteed for solutions of (11) and does not impose further restrictions on the vector-spinor field. To assert the opposite would be analogous to saying that the Klein-Gordon (consistency) condition imposes new restrictions on solutions of the Dirac equation. The Dirac equation in (11) determines, for every $\mu$, half of the $\psi_\mu$ components, while the gamma-trace algebraic constraint removes $k$ components, leaving $\frac{D\times k}{2} - k$ degrees of freedom. In $D = 4$, they amount to 2 spin–$\frac{1}{2}$ and 2 spin–$\frac{3}{2}$ propagating massless helicities. In the massive case, the 4 helicities correspond to $2\times\frac{3}{2} + 1$ polarization states in the spatial rotation irreducible representation [4].

In [15] the Hamiltonian analysis of the massless RS system was carried out, and it was shown that the Dirac algorithm produces two different outputs depending on whether the

---

[2]Here $k = 2^{[D/2]}$ is the number of components of a spinor in $D$ dimensions.

Dirac conjecture—that secondary first-class constraints are gauge symmetry generators—is assumed or not.

In [16], Dirac states: "...it may be that all the first-class secondary constraints should be included among the transformations which don't change the physical state, but I haven't been able to prove it. Also, I haven't found any example for which there exists first-class secondary constraints which do generate a change in the physical state."

Then, Dirac proposes that modifying the field equations by adding the secondary first-class constraints to the total Hamiltonian, thus defining Dirac's extended Hamiltonian $H_E$, new arbitrary velocities appear for those phase space variables which do not change the physical state, and that are equivalent to the first-class Lagrange multipliers associated with the secondary first-class constraints. This procedure renders the evolution of the conjectured unphysical variables also non-deterministic. Thus, adopting Dirac's modified dynamics, the results of the early works [17–20] are reproduced, and the spin–$\frac{1}{2}$ sector is completely removed, a consequence of a second "gauge fixing condition" for the secondary first-class constraints. In the opposite case, the secondary first-class constraints are not regarded as gauge symmetry generators, there are no corresponding gauge fixing conditions to be imposed, and the space of solutions contains the aforementioned propagating Dirac field. The elimination or not of the spin–$\frac{1}{2}$ degree of freedom yields physically different situations, which suffices to say that in this case the Dirac conjecture [16]—that "it may be that all the first-class secondary constraints should be included among the transformations which don't change the physical state"—does not hold.

Complementing [15], here we show that the conclusion that the spin–$\frac{1}{2}$ sector of the massless RS system propagates can also be reached by fixing the gauge in an alternative way, and using the covariant spin-block projectors introduced by Behrends and Fronsdal [21] (see also [1,22]). As we shall see, this result is also valid for supergravity [12,13] and one can write explicitly the spin–$\frac{1}{2}$ content in basic supergravity, consisting of the Dirac action coupled to the background geometry. As a corollary, truncating extended supergravities to their spin–$\frac{1}{2}$ sector must yield a class of unified field theories on curved backgrounds with the coupling constants determined by the supergravity model. This observation was instrumental in building a spin–$\frac{1}{2}$ model of Dirac fermions coupled to a $U(1)$ gauge field and gravitational background starting from a superalgebra in three dimensions [23–26], and in $\mathcal{N}=2$ extensions of Macdowell-Mansouri supergravity [27–29]. Alternatively, this result also implies that spin–$\frac{1}{2}$ states may be generically present in any supergravity theory.

## 2 Unconventional supersymmetry

Its vector-spinor structure shows that the field $\psi_\mu^\alpha$ is in the reducible representation $1 \otimes \frac{1}{2}$ of the Lorentz group. Consider the decomposition of the identity

$$P^{1/2} + P^{3/2} = \mathbb{1}, \quad \text{with} \quad P^{1/2}{}_\mu{}^\nu := \frac{1}{D}\gamma_\mu\gamma^\nu, \quad P^{3/2}{}_\mu{}^\nu := \delta_\mu{}^\nu - \frac{1}{D}\gamma_\mu\gamma^\nu, \tag{13}$$

in terms of spin-block projectors $P^{(s)}P^{(s')} = \delta^s_{s'}P^{(s')}$, $s = \frac{1}{2}, \frac{3}{2}$. Then, the reducible representation $1 \otimes \frac{1}{2}$ can be split into $\frac{3}{2} \oplus \frac{1}{2}$ irreducible components as

$$\psi_\mu := \rho_\mu + \gamma_\mu\kappa, \quad \text{with} \quad \rho_\mu \equiv P^{3/2}{}_\mu{}^\nu\psi_\nu, \quad \text{and} \quad \kappa \equiv \frac{1}{D}\gamma^\mu\psi_\mu, \tag{14}$$

where the spin–$\frac{3}{2}$ part $\rho_\mu$ is gamma-traceless by construction and $\kappa$ is a pure spin–$\frac{1}{2}$ mode. An indication of the non-trivial role of the spin–$\frac{1}{2}$ sector is that the projection of the vector-spinor to the component

$$\tilde{\psi}_\mu = \gamma_\mu\kappa, \tag{15}$$

in the Lagrangian functional (1) reproduces the Dirac term,

$$\mathcal{L}[\tilde{\psi}_\mu = \gamma_\mu \kappa] = i \frac{(D-1)(D-2)}{2} \bar{\kappa} \not{\partial} \kappa \,. \tag{16}$$

This shows that the spin–$\frac{1}{2}$ sector is not in the kernel of the Lagrangian functional. It is not a zero-mode of the action and (16) is not a boundary term. Hence $\kappa$ does not qualify as a "pure gauge" mode. A pure gauge configuration would make the Lagrangian identically vanish, as in the case of Yang-Mills theories, or would reduce it to a total derivative, as in Chern-Simons theories. On the contrary, (16) describes a propagating spin–$\frac{1}{2}$ field that contributes to the energy-stress tensor of the theory.

The result (16), applied to three-dimensional Chern-Simons supergravity produces a spin–$\frac{1}{2}$ model of Dirac fermions coupled to the $U(1)$ gauge field and the gravity background [23, 28, 29] that turns out to be suitable for the description of electrons in graphene-like materials [30–43].

More generally, the projection (15), dubbed *the matter ansatz*, can be used to generate models of spin–$\frac{1}{2}$ fermions coupled to gravity and gauge interactions starting from supersymmetry gauge connections, of the form

$$A_\mu = W_\mu + U_\mu + Q_\alpha \psi_\mu^\alpha \,, \tag{17}$$

where $W$ is valued in a spacetime symmetry group, $U$ in an internal gauge symmetry group, and $Q$ is the supercharge generator. If the matter ansatz is implemented in the fermion sector, the action principle $S[A]$ acquires interactions with a few arbitrary coupling constants, a welcome feature in (grand) unified models. For example, in 3D Chern-Simons supergravity, such action principle is given by $\int \text{str}(A \wedge dA + \frac{2}{3} A \wedge A \wedge A)$. In five-dimensional Chern-Simons supergravity, similar results were found in [44, 45]. The approach of generating spin–$\frac{1}{2}$ systems coupled to gauge and gravity backgrounds using these methods is often referred to as "unconventional supersymmetry" [23–29, 44, 46–52]. Even though these models are not supersymmetric in general, they may have supersymmetric ground states, described by the vanishing curvature constraint $F = dA + A \wedge A = 0$, since under a supersymmetry transformation $\delta F = [F, Q_\alpha \epsilon^\alpha] \equiv 0$.

The general statement that "the spin–$\frac{1}{2}$ mode of the gravitino is pure gauge" seems to be in contradiction with the fact that the matter ansatz applied to the massless RS model (1) gives rise to the Dirac Lagrangian (16) and that the unconventional supersymmetry models describe propagating spin–$\frac{1}{2}$ degrees of freedom. This contradiction is resolved in [15], and here we provide further evidence of the fact that spin–$\frac{1}{2}$ sector of the vector-spinor propagates.

## 3 Covariant analysis

### 3.1 Off-shell gauge fixing

In terms of the decomposition (14) the Lagrangian (1) reads

$$\mathcal{L}[\rho_\mu + \gamma_\mu \kappa] = -\frac{i}{2} \left( \bar{\rho}^\mu \not{\partial} \rho_\mu - (D-1)(D-2) \bar{\kappa} \not{\partial} \kappa + 2(D-2) \bar{\kappa} \partial \cdot \rho \right) , \tag{18}$$

up to boundary term. The Euler-Lagrange equations (9)-(10) now take the form,

$$\not{\partial} \rho_\mu - \gamma_\mu \not{\partial} \kappa - (D-2) \partial_\mu \kappa = 0 \,, \qquad (D-1) \not{\partial} \kappa - \partial^\mu \rho_\mu = 0 \,. \tag{19}$$

We can use the gauge freedom (8) to make $\rho_\mu$ divergence-free in addition to being gamma-traceless. Indeed, consider the gauge transformation $\psi'_\mu = \psi_\mu + \partial_\mu \epsilon$, then

$$\rho'_\mu + \gamma_\mu \kappa' = \rho_\mu + \gamma_\mu \kappa + \partial_\mu \epsilon \,. \tag{20}$$

A spinor field $\epsilon$ can be found such that $\rho'_\mu$ remains gamma-traceless and is also divergence-free,

$$\partial^\mu \rho'_\mu = 0 \, . \tag{21}$$

Taking the gamma trace and the divergence of (20) it is easily checked that the gauge parameter $\epsilon$ that meets these requirements must satisfy the single condition

$$\Box \epsilon = \frac{-D}{D-1} \partial \cdot \rho \, . \tag{22}$$

In this gauge, equations (19) reduce to

$$\not{\partial} \rho'_\mu = (D-2)\partial_\mu \kappa' \, , \qquad \not{\partial} \kappa' = 0 \, . \tag{23}$$

Thus, $\kappa$ is a spin-$\frac{1}{2}$ field that obeys a free massless Dirac equation, while $\rho_\mu$ satisfies the massless Dirac equation with a source—given by $\kappa$—and fulfills all the conditions of a propagating massless spin-$\frac{3}{2}$ field. Thus, in this gauge, $\rho_\mu$ and $\kappa$ propagate massless modes of spin-$\frac{3}{2}$ and spin-$\frac{1}{2}$, respectively.

The result can also be appreciated off-shell. The gauge (21) can be reached from any off-shell configuration (20). For those configurations the Lagrangian (1), in the form (18), takes value

$$\mathcal{L}[\rho'_\mu + \gamma_\mu \kappa'] = -\frac{i}{2}\Big(\bar{\rho}'^\mu \not{\partial} \rho'_\mu - (D-1)(D-2)\bar{\kappa}' \not{\partial} \kappa'\Big) \, , \tag{24}$$

to which the spin-$\frac{1}{2}$ and spin-$\frac{3}{2}$ sectors contribute independently. In the next section, it will become clear that (24) describes the gauge invariant part of the vector spinor field.

## 3.2 Irreducible Poincaré representations

Wigner's classification of particles describes them mathematically as fields in irreducible representations of the Poincaré group. In this case, the vector-spinor $\psi_\mu^\alpha$ can be decomposed in irreducible representations of spins $(1 \oplus 0) \otimes \frac{1}{2} = \frac{3}{2} \oplus \frac{1}{2} \oplus \frac{1}{2}$ [1].

The spin content can be made manifest using spin-block projectors [21]. Following [1], let us consider the operators

$$\hat{\gamma}_\mu = \gamma_\mu - w_\mu \, , \qquad w_\mu = \not{\partial}^{-1}\partial_\mu \, , \tag{25}$$

which span a basis of orthogonal vector operators satisfying

$$\hat{\gamma}_\mu \hat{\gamma}^\mu = (D-1)\mathbb{1} \, , \qquad w_\mu w^\mu = \mathbb{1} \, , \qquad \hat{\gamma}_\mu w^\mu = 0 \, . \tag{26}$$

Hence, the identity operator, acting on vector-spinors, admits the decomposition

$$\mathbb{1} = P^{3/2\,T} + P_{11}^{1/2} + P_{22}^{1/2} \, , \tag{27}$$

$$P_{\mu\nu}^{3/2\,T} = \theta_{\mu\nu} - \frac{1}{D-1}\hat{\gamma}_\mu \hat{\gamma}_\nu \, , \qquad P_{11\,\mu\nu}^{1/2} = \frac{1}{D-1}\hat{\gamma}_\mu \hat{\gamma}_\nu \, , \qquad P_{22\,\mu\nu}^{1/2} = w_\mu w_\nu \, , \tag{28}$$

where

$$\theta_{\mu\nu} := \eta_{\mu\nu} - w_\mu w_\nu \, , \qquad w^\mu \theta_{\mu\nu} = 0 \, , \qquad \hat{\gamma}^\mu \theta_{\mu\nu} = \hat{\gamma}_\nu \, , \qquad \theta_{\mu\nu}\theta^{\nu\lambda} = \delta_\mu^\lambda \mathbb{1} \, . \tag{29}$$

Then, the following identities are verified

$$(\hat{\gamma}^\mu, w^\mu)P_{\mu\nu}^{3/2\,T} = (0,0) \, , \quad (\hat{\gamma}^\mu, w^\mu)P_{11\,\mu\nu}^{1/2} = (\hat{\gamma}_\nu, 0) \, , \quad (\hat{\gamma}^\mu, w^\mu)P_{22\,\mu\nu}^{1/2} = (0, w_\nu) \, . \tag{30}$$

Defining,

$$\rho_\mu^T := P_{\mu\nu}^{3/2\,T}\psi^\nu \, , \tag{31}$$

we see that the traceless-transverse projector $P^{3/2\,T}$ removes the gamma-trace and the divergence of the vector-spinor,

$$\gamma^\mu \rho_\mu^T = 0, \qquad \partial^\mu \rho_\mu^T = 0. \tag{32}$$

The projector $P^{1/2}_{11\,\mu\nu}$ yields a vector along $\hat\gamma$, orthogonal to $w$, and $P^{1/2}_{22\,\mu\nu}$ projects onto $w$:

$$\tilde\kappa = w^\mu \psi_\mu = \frac{\partial^\mu}{\partial\!\!\!/}\psi_\mu, \qquad \kappa = (D-1)^{-1}\hat\gamma^\mu\psi_\mu. \tag{33}$$

The space of vector-spinors $\psi_\mu$ decomposes as

$$\psi_\mu = \rho_\mu^T + \hat\gamma_\mu \kappa + w_\mu \tilde\kappa, \qquad \bar\psi_\mu = \bar\rho_\mu^T - \bar\kappa \hat\gamma_\mu - \bar{\tilde\kappa} w_\mu. \tag{34}$$

The gauge transformation (8) adds to the vector spinor a component, i.e. $\partial_\mu \epsilon$ that is annihilated by the projectors $P^{3/2\,T}_{\mu\nu}$ and $P^{1/2}_{11\,\mu\nu}$, while it belongs to the eigenspace of the longitudinal projector $\partial_\mu \epsilon = P^{1/2}_{22\,\mu}{}^\nu \partial_\nu \epsilon$. It follows that the gauge invariant parts of the vector-spinor consists of $\rho_\mu^T$ and $\kappa$, while the non-invariant part transform as $\delta\tilde\kappa = \partial\!\!\!/ \epsilon$. Introducing the projector operators the RS equations read,

$$\left(P^{3/2}{}_\mu{}^\nu - (D-2)P^{1/2}_{11\,\mu}{}^\nu\right)\partial\!\!\!/\psi_\nu = 0, \tag{35}$$

which, projected onto the Poincaré irreducible spin-$\frac{1}{2}$ and spin-$\frac{3}{2}$ spaces, yield

$$\partial\!\!\!/\rho_\mu^T = 0, \qquad \partial\!\!\!/\kappa = 0, \tag{36}$$

which confirms the redundancy of the longitudinal mode and the physical meaning of $\rho_\mu^T$ and $\kappa$ as propagating degrees of freedom. It is then evident that a suitable gauge fixing condition is simply $\tilde\kappa = 0$, which may then be used in a Gupta-Bleuler quantization scheme.

The gauge transformation (8) affects only the mode along $w_\mu$, and is equivalent to

$$\delta\rho_\mu^T = 0, \qquad \delta\kappa = 0, \qquad \delta\tilde\kappa = \partial\!\!\!/\epsilon, \tag{37}$$

and we have two gauge invariant sectors, $\rho_\mu^T$ and $\kappa$. Substituting (34), the Lagrangian (1) reduces to

$$\mathcal{L}[\rho_\mu^T + w_\mu\tilde\kappa + \hat\gamma_\mu\kappa] = -\frac{i}{2}\left(\bar\rho_\mu^T \partial\!\!\!/ \rho^{T\mu} - (D-1)(D-2)\bar\kappa\partial\!\!\!/\kappa\right), \tag{38}$$

which coincides with (24). We conclude that the RS Lagrangian does not depend on the longitudinal mode $\tilde\kappa$ and the physical degrees of freedom are the gauge invariant ones, $(\rho_\mu^T, \kappa)$.

## 4 Space-time splitting

Splitting the RS field $\psi_\mu$ into its time- and space-like components ($\psi_0$, $\psi_i$; $i = 1, 2, \cdots, D-1$), the Lagrangian (1) reads

$$\mathcal{L} = -i\bar\psi_0 \gamma^{0ij}\partial_i\psi_j + \frac{i}{2}\bar\psi_i\gamma^{0ij}\dot\psi_j - \frac{i}{2}\bar\psi_i\gamma^{ijk}\partial_j\psi_k. \tag{39}$$

The spatial spin-vector $\psi_i$ can be further decomposed by the action of three orthogonal spatial rotation group spin-block projectors, Behrends-Fronsdal's analogs, that add up to the identity. These projectors are

$$(P^N)_{ij} := \frac{1}{D-2}N_i N_j, \qquad (P^L)_{ij} := L_i L_j, \qquad P^T = \mathbb{1} - P^N - P^L, \tag{40}$$

where $N_i := \gamma_i - L_i$ and $L_i := \slashed{\nabla}^{-1}\partial_i$. Then, a spatial vector-spinor splits as

$$\psi_i = \xi_i + N_i\zeta + L_i\lambda\,, \tag{41}$$

where

$$\xi_i = P^T{}_i{}^j\psi_j\,, \qquad \zeta = \frac{1}{D-2}N^i\psi_i\,, \qquad \lambda = L^i\psi_i\,, \tag{42}$$

which can be verified with the help of the identities $N_i N^i = D-2$, $L_i L^i = 1$, $N_i L^i = 0$.

## 4.1 Explicit solution with spin-$\frac{1}{2}$ and spin-$\frac{3}{2}$

The above decomposition of $\psi_\mu$ has not used the gauge symmetry. If one uses the gauge freedom to impose the gamma traceless condition, $\gamma^\mu\psi_\mu = 0$,

$$\psi_0 + \gamma_0\gamma^i\psi_i = 0\,, \tag{43}$$
$$\psi_0 + \gamma_0((D-2)\zeta + \lambda) = 0\,, \tag{44}$$

implies that $\psi_0$ is not an independent field.

Together with decomposition (41), the field equations (9) reduce to

$$\slashed{\partial}\xi_i = 0\,, \qquad \slashed{\partial}\tilde{\lambda} = 0\,, \qquad \dot{\zeta} = 0\,, \qquad \slashed{\nabla}\zeta \approx 0\,, \tag{45}$$

where $\tilde{\lambda} = \gamma^0\lambda$. From the last two equations, it follows that the auxiliary spinor $\zeta$ is constant and it cannot be normalized. Thus, we find the explicit solutions to the field equations (9)

$$\zeta = 0\,, \qquad \psi_0 = \gamma^0\lambda\,, \qquad \psi_i = \xi_i + \partial_i\slashed{\nabla}^{-1}\lambda\,, \tag{46}$$

where $\tilde{\lambda}$ and $\xi_i$ satisfy the standard Dirac equations (45), and $\xi_i$ is standard double-transverse solution found in [19, 53] for gauge fixed RS equations. It follows that $\xi_i$ and $\lambda$ propagate massless fields of spin–$\frac{3}{2}$ and spin–$\frac{1}{2}$, respectively. Upon (46) the Pauli-Lubanski pseudo-vector yields the standard relation $W_\mu\psi = \mathfrak{s}P_\mu\psi$, $\mathfrak{s} = \pm\frac{3}{2}, \pm\frac{1}{2}$ respectively for the chiral projections, $\frac{1\pm\gamma_5}{2}\xi_i$ and $\frac{1\pm\gamma_5}{2}\partial_i\slashed{\nabla}^{-1}\lambda$.

Since no arbitrary functions of time remain in the system, the dynamical equations (45) ultimately determine the evolution of the fields from initial data on a Cauchy surface, and there is no need for additional gauge fixing conditions.

## 5 Hamiltonian analysis

The previous discussion establishes the degrees of freedom of the RS system from the Lagrangian equations of motion in a particular gauge. The time-honored constrained Hamiltonian formalism of Dirac [16, 54, 55] provides an independent and reliable test for the consistency of this result.

Dirac's algorithm starts from the Hamiltonian, separating spacetime and fields into their temporal and spatial components (39), and carrying out the Legendre transform defining the canonical momenta, $\pi^\mu := \partial\mathcal{L}/\partial\dot{\psi}_\mu$. In the case at hand, this yields the primary constraints

$$\pi^0 \approx 0\,, \tag{47}$$

$$\chi^i := \pi^i - \frac{i}{2}\mathcal{C}^{ij}\psi_j \approx 0\,, \qquad \{\chi^i, \chi^j\} = i\mathcal{C}^{ij}\,. \tag{48}$$

Since the matrix $\mathcal{C}^{ij}_{\alpha\beta} := -(C\gamma^{0ij})_{\alpha\beta} = \mathcal{C}^{ji}_{\beta\alpha}$ is invertible, $\mathcal{C}^{ij}_{\alpha\beta}(\mathcal{C}^{-1})^{\beta\kappa}_{jm} := \delta^i_m \delta^\kappa_\alpha$,

$$(\mathcal{C}^{-1})^{\alpha\beta}_{ij} = \Big(-\frac{1}{(D-2)}\gamma_i\gamma_j\gamma_0 C^{-1} + \delta_{ij}\gamma_0 C^{-1}\Big)^{\alpha\beta}, \tag{49}$$

and consequently, the constraints $\chi^i$ are second-class. The primary constraints (47, 48) result from the non-dynamical nature of $\psi_0$ and the first-order character of the system, respectively. The *total Hamiltonian* includes the Canonical Hamiltonian and a linear combination of the primary constraints,

$$H_T = \int d^{D-1}x\, \Big(i\bar{\psi}_0\gamma^{0ij}\partial_i\psi_j + \frac{i}{2}\bar{\psi}_i\gamma^{ijk}\partial_j\psi_k + \chi^i_\alpha\mu^\alpha_i + \pi^0_\alpha\mu^\alpha_0\Big), \tag{50}$$

where $\mu_i$ and $\mu_0$ are Lagrange multipliers. Preservation in time of the constraint $\pi^0 \approx 0$ yields a secondary constraint,

$$\dot{\pi}^0 = -\frac{\delta H_T}{\delta\psi_0} = -iC\gamma^{0ij}\partial_i\psi_j \approx 0 \qquad \Leftrightarrow \qquad \varphi := -i\mathcal{C}^{ij}\partial_i\psi_j \approx 0, \tag{51}$$

which is the field equation obtained by varying (39) with respect to $\psi_0$. Here, the Poisson bracket is given by

$$\{f(t,\vec{x}), g(t,\vec{y})\} := (-1)^{|f|}\int d^{D-1}z\, \Big(\frac{\delta f(t,\vec{x})}{\delta\psi^\alpha_\mu(t,\vec{z})}\frac{\delta g(t,\vec{y})}{\delta\pi^\mu_\alpha(t,\vec{z})} + \frac{\delta f(t,\vec{x})}{\delta\pi^\mu_\alpha(t,\vec{z})}\frac{\delta g(t,\vec{y})}{\delta\psi^\alpha_\mu(t,\vec{z})}\Big). \tag{52}$$

Demanding preservation in time of the second-class constraint $\chi^i \approx 0$ yields

$$\dot{\chi}^i = -(C\gamma^i\gamma_0 C^{-1})\varphi + i\mathcal{C}^{ij}\partial_j\psi^\beta_0 + iC\partial^i\gamma^j\psi_j - iC\slashed{\partial}\psi^i + i\mathcal{C}^{ij}\mu_j \approx 0, \tag{53}$$

which determines $\mu^i$ in terms of the canonical variables and introduces no new constraints. The preservation of the secondary constraint $\varphi$ yields conditions on part of the second-class constraint Lagrange multipliers $\mu^i$,

$$\dot{\varphi} = i\mathcal{C}^{ij}\partial_i\mu_j = -C\gamma^0\slashed{\partial}N^j\mu_j \approx 0. \tag{54}$$

This is equivalent to setting $N_i\mu^i = 0$ in a decomposition analogous to (41), and is consistent with the stationary condition (53) on the primary second-class constraints. Thus, the preservation in time of $\varphi \approx 0$ implies the vanishing of the component of $\mu_i$ along $N^i$. Since first-class constraints do not determine Lagrange multipliers, this shows that $\varphi$ is not a first-class generator. Indeed, it can be easily checked that the linear combination

$$\tilde{\varphi} := \varphi + i\partial_i\chi^i \approx 0, \tag{55}$$

is first-class. It follows that the secondary constraint $\varphi$ is a linear combination of first-class and second-class constraints.

The second-class constraints can be eliminated by setting $\chi^i$ strongly to zero, reducing the system to the surface of the second-class constraints and replacing Poisson brackets with Dirac ones, $\{f,g\}_D := \{f,g\} - \{f,\chi^i_\alpha\}\mathcal{C}^{-1\,\alpha\beta}_{ij}\{\chi^j_\beta,g\}$,

$$\{f,g\}_D = (-1)^f\int d^{D-1}z\,\Big[-i\frac{\delta f}{\delta\xi_i}P^T_{ij}\gamma_0 C^{-1}\frac{\delta g}{\delta\xi_j} - i\frac{D-3}{D-2}\frac{\delta f}{\delta\lambda}\gamma_0 C^{-1}\frac{\delta g}{\delta\lambda}$$
$$+i\frac{1}{D-2}\Big(\frac{\delta f}{\delta\lambda}\gamma_0 C^{-1}\frac{\delta g}{\delta\zeta} + \frac{\delta f}{\delta\zeta}\gamma_0 C^{-1}\frac{\delta g}{\delta\lambda}\Big) + \Big(\frac{\delta f}{\delta\psi^\alpha_0}\frac{\delta g}{\delta\pi^0_\alpha} + \frac{\delta f}{\delta\pi^0_\alpha}\frac{\delta g}{\delta\psi^\alpha_0}\Big)\Big]. \tag{56}$$

The first-class Hamiltonian (50) in the reduced phase space becomes

$$H_R = \int d^{D-1}x \left( i(D-2)\bar{\psi}_0\gamma^0\slashed{\nabla}\zeta - \frac{i(D-2)(D-3)}{2}\bar{\zeta}\slashed{\nabla}\zeta + \frac{i}{2}\bar{\xi}^i\slashed{\nabla}\xi_i + \pi^0_\alpha\mu^\alpha_0 \right), \qquad (57)$$

where the secondary first-class constraint $\varphi \approx 0$ is equivalent to $\slashed{\nabla}\zeta \approx 0$. Thus, we arrive at a constrained Hamiltonian system for the reduced phase space where $\pi^0 \approx 0$ and $\varphi \approx 0$ are the only remaining first-class constraints.

## 5.1 Consequences of the constraint $\varphi \approx 0$

Up to this point, our analysis agrees with the Hamiltonian analysis in references [17–20]. Henceforth, two paths can be followed depending on whether the *Dirac conjecture*[3] (**DC**) is adopted or not, with the two options resulting in distinct physical systems, *i.e.* with different numbers of propagating degrees of freedom.

The action principle,

$$S = \int dt(\dot{\psi}_\mu\pi^\mu - H_T), \qquad (58)$$

given in terms of the total Hamiltonian (50), yields field equations equivalent to the Euler-Lagrange equations. This is because only primary constraints are necessary to recover the starting Lagrangian.

In Dirac's extended dynamics, secondary first-class constraints are considered independent gauge symmetry generators and are added to the Hamiltonian with their corresponding Lagrange multipliers. In this case the *extended Hamiltonian* reads,

$$H_E := H_T + \tau^\alpha\tilde{\varphi}_\alpha, \qquad (59)$$

where $\tau$ is a new Lagrange multiplier. In this framework, gauge fixing conditions should be imposed to intersect the gauge orbits generated by the $\tau^\alpha\tilde{\varphi}_\alpha$ component and determine $\tau$.

On the surface of second-class constraint $\chi_i = 0$, the Hamiltonian reduces to

$$\tilde{H}_E := H_R + \tau^\alpha\varphi_\alpha. \qquad (60)$$

Then, the evolution defined by the (reduced) extended Hamiltonian (60) is given by,

$$\dot{\xi}_i = -\gamma_0\slashed{\nabla}\xi_i, \qquad \dot{\lambda} = -(D-3)\gamma_0\slashed{\nabla}\zeta + \slashed{\nabla}\psi_0 + \slashed{\nabla}\tau, \qquad (61)$$

$$\dot{\psi}_0 = -\mu_0, \qquad \pi^0 = 0, \qquad \zeta = 0, \qquad \slashed{\nabla}\zeta = 0. \qquad (62)$$

The gauge orbits generated by $\mu_0\pi^0$ are intersected by a surface defined by fixing $\psi_0$, which can be chosen to implement the standard gamma-traceless condition (43)-(44). This and the constraint $\pi^0 \approx 0$ form a second-class pair. Hence, the Lagrange multiplier $\mu_0$ can be eliminated by demanding (44) to be stationary, relating $\mu_0$ to $\lambda$ and to the Lagrange multiplier $\tau$,

$$\mu_0 \approx \slashed{\nabla}\lambda - \gamma_0\slashed{\nabla}\tau. \qquad (63)$$

The gauge choice (44) is accessible since $\pi^0$ generates arbitrary shifts in $\psi_0$. In particular the shift $\delta\psi_0 = -(\psi_0 + \gamma_0\gamma^i\psi_i)$ renders $\psi_\mu$ gamma-traceless. Thus in the phase space spanned by $\xi_i, \zeta, \lambda$, the system (61, 62) reduces to

$$\dot{\xi}_i + \gamma_0\slashed{\nabla}\xi_i = \gamma_0\slashed{\partial}\xi_i = 0, \qquad \dot{\lambda} - \gamma_0\slashed{\nabla}\lambda = \slashed{\nabla}\tau, \qquad \zeta = 0 = \slashed{\nabla}\zeta. \qquad (64)$$

---

[3]At the end of Chapter 1 of [16], Dirac conjectured that all first-class constraints generate gauge symmetries.

Here $\tau$ is the only arbitrary function of time remaining in the system. In order to eliminate $\tau$, an external gauge fixing condition conjugate to $\varphi \approx 0$ becomes necessary. Since $\lambda$ is conjugate to the constraint sourced by $\tau$, it must be gauge-fixed, and nothing can prevent its removal, leaving $\xi_i$ as the only propagating field in the system. This is how the spin-$\frac{1}{2}$ mode is removed in [17–20].

## 5.2 Consequences of Dirac's conjecture

Some comments are in order here. First, the Euler-Lagrange equations (45) match the Hamiltonian evolution equations obtained from (64) only if $\tau = 0$. Second, the equation for $\lambda$ in (64) reads

$$\partial\!\!\!/\,\tilde{\lambda} = \nabla\!\!\!\!/\,\tau\,, \qquad \tilde{\lambda} := \gamma_0 \lambda\,, \tag{65}$$

and it has the form of an inhomogeneous Dirac equation. The solution of this equation can be expressed as the sum of a homogeneous plus an inhomogeneous part, $\tilde{\lambda} = \tilde{\lambda}_{ho} + \tilde{\lambda}_{in}$, such that

$$\partial\!\!\!/\,\tilde{\lambda}_{ho} = 0\,, \qquad \tilde{\lambda}_{in} = \partial\!\!\!/^{-1}\nabla\!\!\!\!/\,\tau\,, \tag{66}$$

where $\partial\!\!\!/^{-1}$ is the Green function for the Dirac operator. Thus $\lambda = \gamma^0 \tilde{\lambda}$ contains a part with indeterminate time evolution ($\tilde{\lambda}_{in}$) that depends on the arbitrary Lagrange multiplier $\tau$, and the homogeneous part $\tilde{\lambda}_{ho}$ whose time evolution is entirely deterministic. We conclude that the arbitrary time dependence in the spin-$\frac{1}{2}$ field comes from the "human input" $\tau$, and is, therefore, an artifact of the procedure brought about by the insistence on assuming $\varphi$ as a gauge generator.

This shows that the DC is not logically necessary and therefore, $\varphi$ need not be regarded as a gauge generator to be added to $H_T$. Dropping $\varphi$ from the set of gauge generators is equivalent of setting $\tau = 0$ in (64) and, consequently, allowing the Hamiltonian field equations to match the Lagrangian ones (45). With $\tau = 0$, the Lagrange multiplier $\mu_0$ is completely determined from (63), as expected, and $\tilde{\lambda} = \tilde{\lambda}_{ho}$ propagates as a standard Dirac field, thus recovering solution (46).[4]

## 5.3 Unified models from supergravity

The result (16) can be extended to supergravity. Assuming the matter Ansatz (15) –and allowing $\gamma^\mu \psi_\mu \neq 0$–, the spin-$\frac{3}{2}$ sector is completely removed. This produces a theory in which the fermionic sector is purely made of spin-$\frac{1}{2}$ fermions coupled to gravity and gauge fields. For example, consider the *basic supergravity* model [12, 13] given by

$$\mathcal{L}_{\text{sugra}}[e_\mu^a, \omega_\mu^{ab}, \psi_\mu] = \mathcal{L}_{\text{RS}} + \frac{1}{2}e R_{\mu\nu}^{ab} e_a^\mu e_b^\nu\,, \tag{68}$$

where $e_a^\mu$ is the inverse vielbein. A straightforward computation shows that this Lagrangian restricted to the sector $\psi_\mu = \gamma_\mu \kappa$ reduces to

$$\mathcal{L}_{\text{U-sugra}}[e_\mu^a, \omega_\mu^{ab}, \gamma_\mu \kappa] = \frac{i(D-1)(D-2)}{2}e\,\bar{\kappa}\partial\!\!\!/\kappa + \frac{i(D-1)}{2}\bar{\kappa}\,\gamma^{\mu\nu}T_{\mu\nu}^a\gamma_a\,\kappa + \frac{1}{2}e R_{\mu\nu}^{ab} e_a^\mu e_b^\nu\,, \tag{69}$$

---

[4]The reader can verify that $\varphi$ does not generate an independent symmetry of the field equations induced by the total Hamiltonian, equivalent to (61)-(62) with $\tau = 0$. The true gauge generator corresponds to the Castellani chain [56],

$$G := \int d^{D-1}x(\pi^0\dot{\epsilon} - \bar{\epsilon}\gamma_0\nabla\!\!\!\!/\zeta)\,. \tag{67}$$

More generally, each independent gauge orbit corresponds to a different Castellani chain, naturally superseding the DC [56–58].

where $e = det[e^a_\mu]$, and $T^a_{\mu\nu} = \partial_\mu e^a_\nu + \omega^a{}_{b\mu} e^b_\nu - (\mu \leftrightarrow \nu)$ is the torsion.

A Lagrangian of the form (69) is contained in every extension of most supergravity models [12, 13], including AdS supergravity [59–61], extensions with auxiliary fields [62–65], interactions with gauge [66–68] and matter [69–73], as well as in the superspace formulation of SUGRA [74–78]. The resulting unified theories will be reviewed in a forthcoming article.

# 6 Discussion and Summary

An argument usually used to discard the spin–$\frac{1}{2}$ component is attributed to a "residual" gauge symmetry, with a gauge parameter satisfying the Dirac equation (see *e.g.* [14]). However, shifting the solution of the Dirac equation by a parameter field that also satisfies the Dirac equation merely results in a trivial redefinition of the integration constants. On the other hand, demanding that residual symmetry leaves the initial conditions invariant would force the gauge parameter to vanish. Note that this sort of "symmetry" would be present in any linear field equation, but this cannot be invoked to say that every field satisfying a linear equation could be gauged away.

The use of the decomposition of the vector-spinor gauge field in Poincaré group irreducible components (34), $\rho^T_\mu + \hat{\gamma}_\mu \kappa + w_\mu \tilde{\kappa}$, in the RS Lagrangian reveals the off-shell gauge invariant components of the massless RS system, as shown in (38), consisting the spin–$\frac{3}{2}$ component $\rho^T_\mu$, and spin–$\frac{1}{2}$ $\kappa$. The analog approach to Maxwell's theory would split the gauge field into transverse and longitudinal components, $A_\mu = A^T_\mu + A^L_\mu$, reducing the Lagrangian to $-\frac{1}{4} F^2 = A^T_\mu \Box A^{T\,\mu}$, which depends only on the gauge invariant transverse mode. Note that using the explicit expression for the transverse projector $P^T_{\mu\nu} := \eta_{\mu\nu} - \Box^{-1} \partial_\mu \partial_\nu$, the field equation $\Box A^T_\mu = \Box P^T_{\mu\nu} A^\nu = 0$ is identical to Maxwell's equation. Although we do not normally work with these expressions, the differential projection operators are useful to identify the field components that belong to the kernel of the action functional and those that contribute non-trivially to the action and propagate, as in (36).

We have argued that the massless vector-spinor system usually considered in supergravity, referred to as massless RS system, describes not only spin–$\frac{3}{2}$ degrees but also spin–$\frac{1}{2}$. We have shown this by four alternative methods. In the first approach, we fix the gauge to make the separation of the spin–$\frac{1}{2}$ sector manifest, described by the standard Dirac equation. In the second approach, spin-block projectors are employed to decouple the gauge invariant and the pure gauge mode of the vector-spinor. The pure gauge mode is in the kernel of the Lagrangian functional, while two gauge-invariant sectors remain and propagate as spin–$\frac{1}{2}$ and spin–$\frac{3}{2}$ massless fields, respectively. In the third approach, we split the vector spinor into time-like and space-like vector-spinor components and build an explicit solution of the RS equation containing spin–$\frac{1}{2}$ and spin–$\frac{3}{2}$ sectors.

Our results are compatible with reference [79], where Heidenreich demonstrates that within the quantum field theory of the massless Rarita-Schwinger one of the spin–$\frac{1}{2}$ mode is associated with states possessing 0-norm (the pure-gauge mode), while the spin–$\frac{1}{2}$ and spin–$\frac{3}{2}$ sectors have positive norms.

Following Dirac's Hamiltonian analysis, we arrive at the same results if the Dirac conjecture is not assumed in the formalism. Indeed, the RS system (1) could be seen as a fermionic counterexample to the DC, complementing the bosonic cases found in, *e.g.*, [55, 80, 81]. Postulating the Dirac conjecture to be valid, *a priori*, yields a system with no inconsistencies. However, the Hamiltonian and Euler-Lagrange equations would not be equivalent, and the spin–$\frac{1}{2}$ propagating degrees of freedom would be lost as in [17–20, 53]. In the counterexamples to the DC, considered in [55], it is claimed that the reduction under secondary first-class

constraint without gauge fixing conditions leads to odd-dimensional reduced phase spaces, where the Dirac bracket is ill-defined, preventing its quantization. Consequently, it is proposed to assume the validity of the DC as a general rule [55]. However, it has been shown that quantizing first and then restricting to the surface of secondary first-class constraints avoids these obstructions [82]. Furthermore, in fermionic systems, such as the one that concerns us here, the Dirac bracket is symmetric in Grassmann-odd variables and the constraints are self-conjugate [83].

We also conclude that basic supergravity reduces in spin–$\frac{1}{2}$ sector of the vector-spinor to a coupled Einstein-Dirac system, including torsion terms (69). More generally, in extended supergravity, the spin–$\frac{1}{2}$ sectors will inherit the coupling constants of the gravitational, gauge, and matter interactions, and the spin–$\frac{1}{2}$ projection of these supergravities are grand unification models including gravity. These results encourage exploring this missed supergravity sector as unification candidates. This observation also applies to the group theoretical approach of supergravity [84–87], and to higher dimensional Chern-Simons supergravity [88, 89].

## Acknowledgments

We thank L. Andrianopoli, L. Castellani, R. Noris and M. Trigiante, for enlightening discussions and helpful suggestions.

**Funding information**    This work was partially funded by ANID/FONDECYT grants 1201208, 1220862 and 1230112, and USS-VRID project VRID-INTER22/10.

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
