# Peer review of "Massless Rarita-Schwinger equations: half and three halves spin solution"

_SciPost Physics, doi:SciPost Phys. 16, 065 (2024)_

## Round 1 · Referee Report · Anonymous (Referee 1) · 2023-11-14

Strengths

The content is mathematically well-founded

The results merit recognition

Weaknesses

Unnecessary repetitions

Excessive details that hinder clear understanding.

Report

In the submitted paper, the authors continue their exploration of a two-parameter ($M$, $A$) family of Rarita-Schwinger (RS) Lagrangians, which they initially introduced in a previous work (Ref. 15). Their primary motivation stems from supergravity, with a particular focus on the massless limit, gauge symmetries, and revealing the true tensor-spin characteristics of the resulting first-order linear differential system.

In this paper, they delve deeper into their findings from Ref. 15, particularly addressing the apparent contradiction between the statement that ``the spin 1/2 mode of the gravitino is pure gauge'' and the observation that applying the matter ansatz to the massless Rarita-Schwinger model results in the Dirac term within the Lagrangian functional. Additionally, they discuss the unconventional supersymmetry models that describe propagating spin-1/2 degrees of freedom. The authors provide further evidence supporting the idea that the spin-1/2 sector of the vector-spinor propagates.

Throughout their analysis, they emphasize the significance of Poincaré symmetry in the context of the vector-spinor RS field. They employ the Dirac formalism of constrained Hamiltonian to establish the consistency of their results.

I believe the content is mathematically well-founded. The results merit recognition, even in light of the current absence of robust empirical validation for the supergravity theory.

I have several suggestions that can enhance the quality of the submission's content.

1) Dirac Conjecture Reminder: To provide context and clarity, the authors could start the paper by reminding the reader of Dirac's conjecture. This will set the stage for the subsequent discussion and results in the paper.

2) Clarity and Technical Details: It's essential to maintain clarity throughout the paper. To address this concern, please consider the following points: 2.1 Streamline the presentation by avoiding unnecessary repetitions and providing clear references to earlier sections or equations where appropriate. 2.2 Use subheadings to break down the paper into more digestible sections, making it easier for the reader to navigate through the technical details. 2.3 Whenever possible, use intuitive explanations or examples to illustrate complex concepts or mathematical derivations.

3) Organizing Solutions: To enhance clarity and distinguish between the physically relevant sector and gauge sectors, consider the following improvements:

3.1 Organize the space of solutions in a way that clearly identifies invariant subspaces under the Poincaré group or Lie algebra representations. 3.2 Explicitly highlight and discuss the physical sector, while addressing the emergence of gauge sectors (such as Gupta-Bleuler structures in the massless case).

Incorporating these suggestions should help enhance the clarity and organization of the paper, making it more accessible to the readers while preserving the mathematical rigor of the content.

Requested changes

1) Dirac Conjecture Reminder: To provide context and clarity, the authors could start the paper by reminding the reader of Dirac's conjecture. This will set the stage for the subsequent discussion and results in the paper.

2) Clarity and Technical Details: It's essential to maintain clarity throughout the paper. To address this concern, please consider the following points: 2.1 Streamline the presentation by avoiding unnecessary repetitions and providing clear references to earlier sections or equations where appropriate. 2.2 Use subheadings to break down the paper into more digestible sections, making it easier for the reader to navigate through the technical details. 2.3 Whenever possible, use intuitive explanations or examples to illustrate complex concepts or mathematical derivations.

3) Organizing Solutions: To enhance clarity and distinguish between the physically relevant sector and gauge sectors, consider the following improvements:

3.1 Organize the space of solutions in a way that clearly identifies invariant subspaces under the Poincaré group or Lie algebra representations. 3.2 Explicitly highlight and discuss the physical sector, while addressing the emergence of gauge sectors (such as Gupta-Bleuler structures in the massless case).

  • validity: high
  • significance: high
  • originality: top
  • clarity: good
  • formatting: good
  • grammar: good

Author:  Mauricio Valenzuela  on 2023-12-06  [id 4174]

(in reply to Report 1 on 2023-11-14)

Dear Editor,

We are grateful for the referee's time and effort in carefully reading and evaluating our manuscript. We appreciate the review and its constructive feedback.

We have addressed the referee's comments and made substantial revisions to enhance the clarity and coherence of our manuscript. We have provided a comprehensive outline of the modifications we made in response to each of the referee's requests.

Request 1:
The referee suggested reminding the reader of Dirac's conjecture. We have incorporated this suggestion, integrating two new paragraphs. The first quotes the Dirac conjecture from his "Introduction to Quantum Mechanics" book (ref.20) and the second provides further details.

Request 2:
To address the second request, we have relocated two paragraphs—one from the introduction (end of page 4) and another from the end of section 3—to the newly created section 9, "Discussion and Summary" (formerly "Summary"), as the first two paragraphs. This restructuring enhances the logical flow of the paper.

We have also introduced new subsections, 4.1 and 5.2, to break down the text into smaller, more digestible pieces, enhancing the overall readability of the manuscript.

Request 3:
In response to the third request, a new paragraph has been inserted after equation (34), where we provide a detailed explanation of the invariant subspaces of the vector-spinor field and their correspondence to the Poincaré group. This allows for clearer identification of the pure-gauge component of the field content, making it suitable for use as a Gupta-Bleuler structure.

Furthermore, we have introduced a new subsection, 4.1, titled "Explicit Solution with Spin–1/2 and Spin–3/2," to underscore that this is one of the paper's main results.

The additions and modifications are highlighted in blue for easy identification.

We believe that these changes address the referee's concerns and significantly enhance the quality of our manuscript. We sincerely hope that these revisions meet the expectations of the referee and the editorial.

We thank the opportunity to revise and resubmit our manuscript. We look forward to your favorable consideration.

Sincerely,
The authors.

Attachment:

SciPostPhysicsV2.pdf

---

## Round 1 · Referee Report · Anonymous (Referee 2) · 2024-2-2

Strengths

  1. Nicely reviews the RS Lagrangian and some of its shortcomings.

  2. Shows in a number of ways that the spin-1/2 sector propagates, on top of the 3/2-sector.

Weaknesses

  1. Does not clarify how the propagation of the spin-1/2 sector would modify the usual matching of bosoninc vs fermionic degrees of freedom in supergravity.

Report

The manuscript addresses a usually overlooked aspect of the Rarita-Schwinger action. It provides a very clear analysis and arguments as to why the spin-1/2 degrees of freedom propagate.

---

## Editorial Decision

published